# Kelpwatch: A new visualization and analysis tool to explore kelp canopy dynamics reveals variable response to and recovery from marine heatwaves

Tom W. Bell[1]*, Kyle C. Cavanaugh[2], Vienna R. Saccomanno[3], Katherine C. Cavanaugh[2], Henry F. Houskeeper[1,2], Norah Eddy[3], Falk Schuetzenmeister[3], Nathaniel Rindlaub[3], Mary Gleason[3]

1 Department of Applied Ocean Physics and Engineering, Woods Hole Oceanographic Institution, Woods Hole, Massachusetts, United States of America, 2 Department of Geography, University of California Los Angeles, Los Angeles, California, United States of America, 3 The Nature Conservancy, Sacramento, California, United States of America

* tbell@whoi.edu

**Data Availability Statement:** All data files are available from the Environmental Data Initiative database (doi:10.6073/pasta/

## Abstract

Giant kelp and bull kelp forests are increasingly at risk from marine heatwave events, herbivore outbreaks, and the loss or alterations in the behavior of key herbivore predators. The dynamic floating canopy of these kelps is well-suited to study via satellite imagery, which provides high temporal and spatial resolution data of floating kelp canopy across the western United States and Mexico. However, the size and complexity of the satellite image dataset has made ecological analysis difficult for scientists and managers. To increase accessibility of this rich dataset, we created Kelpwatch, a web-based visualization and analysis tool. This tool allows researchers and managers to quantify kelp forest change in response to disturbances, assess historical trends, and allow for effective and actionable kelp forest management. Here, we demonstrate how Kelpwatch can be used to analyze long-term trends in kelp canopy across regions, quantify spatial variability in the response to and recovery from the 2014 to 2016 marine heatwave events, and provide a local analysis of kelp canopy status around the Monterey Peninsula, California. We found that 18.6% of regional sites displayed a significant trend in kelp canopy area over the past 38 years and that there was a latitudinal response to heatwave events for each kelp species. The recovery from heatwave events was more variable across space, with some local areas like Bahía Tortugas in Baja California Sur showing high recovery while kelp canopies around the Monterey Peninsula continued a slow decline and patchy recovery compared to the rest of the Central California region. Kelpwatch provides near real time spatial data and analysis support and makes complex earth observation data actionable for scientists and managers, which can help identify areas for research, monitoring, and management efforts.

93b47266b20bc1782c8df9c36169e372). These data are also cited in the manuscript (Bell et al. 2022).

**Funding:** TB was supported by The Nature Conservancy grant (P119034; https://www.nature.org/). FS and NR are employees of The Nature Conservancy and developed the backend and frontend of the Kelpwatch.org website and provided reviews of the manuscript. VS, NE, and MG are employees of The Nature Conservancy and provided supervision of the Kelpwatch.org website development and reviews of the manuscript. TB and KC were funded by the National Aeronautics and Space Administration Ocean Biology and Biogeochemistry grant (80NSSC21K1429; https://www.nasa.gov/). KC was funded by the National Science Foundation Division of Ocean Sciences grant (1831937; https://www.nsf.gov/div/index.jsp?div=OCE).

**Competing interests:** The authors have declared that no competing interests exist.

## Introduction

Along the west coast of North America, underwater forests of kelp provide the foundation for a productive and diverse nearshore ecosystem [1]. The dominant and iconic species of kelp in this region are giant kelp (*Macrocystis pyrifera*) and bull kelp (*Nereocystis luetkeana*), both of which create large, floating canopies. Both species have high rates of primary production [2] and create complex structure [3], thereby providing food and habitat for many ecologically and economically important species. However, the abundance of these kelp species fluctuates rapidly and is sensitive to environmental changes [4]. Stressors such as climate change, overgrazing, and coastal development have been linked to declines in kelp abundance [5] and there is high spatial variability in the response of kelp forests to changing environmental conditions [6].

From 2014 to 2016 the west coast of North America experienced a series of extreme marine heatwaves that had significant impacts to coastal marine ecosystems [7, 8] and was the warmest three-year period on record for the California Current [9]. This heatwave period initially led to widespread declines in the abundance of giant and bull kelp [10, 11], but the magnitude and duration of these impacts varied widely. In northern California, the combined effects of the heatwaves, the loss of an important sea urchin predator (sunflower sea stars) due to disease [12], and a subsequent explosion in sea urchin populations led to a collapse in bull kelp abundance, with devastating ecological and economic impacts [11, 13]. However, despite the regional loss of sunflower sea stars [12], bull kelp populations in southern Oregon were relatively insensitive to the heatwave events [14]. Around the Monterey Peninsula in Central California, increased sea urchin abundance has reduced the once expansive giant kelp forests to a patchwork of urchin barrens and kelp stands that are maintained by sea otters (an important sea urchin predator) selectively feeding on healthy urchins within the remaining kelp areas [15]. In southern California and across the Baja California Peninsula there were widespread declines in giant kelp abundance immediately following the heatwave events, but recovery in subsequent years was spatially variable [10].

Frequent and widespread monitoring of kelp forests is crucial for understanding patterns and drivers of kelp forest trends and their response to disturbances, which is a key component of effective kelp forest ecosystem-based management [16]. Many species of kelp (including bull kelp and giant kelp) have populations that are highly variable through time [17, 18]. Boom and bust cycles are common, and collapse of kelp forests can be sudden [13, 19]. Kelp forest dynamics are also highly variable on small spatial scales (e.g., kilometers, [20]), which leads to high amounts of variability in patterns of recovery, even following widespread disturbance events such as continental-scale marine heatwaves [10, 21].

Remote sensing is a powerful tool for monitoring canopy forming kelps such as bull kelp and giant kelp, and recent increases in the availability of airborne and spaceborne imagery is enabling regular monitoring across multiple space and time scales [5, 14, 22]. For example, inexpensive small unoccupied aerial systems (UAS) can provide very high-resolution monitoring of canopy extent at local scales [23, 24], constellations of CubeSats can provide high-resolution data on regional scales [25], while moderate resolution satellites can be used to map kelp canopy dynamics at global scales [26, 27]. The Landsat satellite program is particularly valuable for kelp monitoring, as it provides imagery with continuous global coverage at a 30 m resolution since 1984, and can be used to detect long-term trends in kelp canopy area, biomass, and abundance and put recent changes in a broader historical context [18, 20].

Landsat imagery has been used to map both giant kelp and bull kelp canopy density and extent [14, 18, 28–30] and kelp abundance [18, 20] on seasonal time scales from 1984 to present for the west coast of the United States and Baja California, Mexico [14, 18, 28–30] and

other regions of the world [27, 31, 32]. One of the most valuable aspects of this dataset is its extensive spatial and temporal coverage, especially for distinguishing the impacts of climate change on kelp populations from other sources of variability [5]. However, the size of the dataset also makes it difficult to use, especially for those without extensive experience working with large geospatial datasets and more complicated file formats. This accessibility barrier has limited the use of the Landsat dataset for mapping and monitoring canopy forming kelps.

To increase accessibility of Landsat imagery among researchers, management agencies, and the public, we created Kelpwatch.org, a visualization and analysis web tool that allows users to select a region, time frame, and season(s) of interest to interactively display changes in kelp canopy over time and freely download data. The primary objective of Kelpwatch.org is to make published kelp canopy data from Landsat imagery actionable for restoration practitioners and researchers, and promote data-driven resource management (e.g., targeted restoration efforts, adaptively managing kelp harvest leases, changing fisheries seasons or catch limits). Analogous web tools have demonstrated success in facilitating data-driven management of other foundational ecosystems by making earth observation data actionable (e.g., Global Forest Watch, Allen Coral Atlas, Global Mangrove Watch; [33–36]).

Kelpwatch.org (hereafter referred to as Kelpwatch) provides a user-friendly interface to analyze and download seasonal kelp canopy observations at 30 m resolution for the west coast of North America from central Baja California, Mexico to the Washington-Oregon border since 1984. To demonstrate the types of analyses that can be completed using Kelpwatch, we used data downloaded directly from Kelpwatch to ask the following questions: (1) What were the regional trends in kelp canopy area over the past 38 years? (2) What were the spatial patterns of kelp canopy area in response to and recovery from the 2014 to 2016 marine heatwave events? and (3) Given the recent spatial alterations to kelp forests by sea urchins around the Monterey Peninsula, California [15], how do local-scale patterns in recent kelp canopy area in this subregion compare to historical data?

## Methods

### Kelpwatch platform

We developed Kelpwatch to make the Landsat kelp canopy dataset actionable via a user-friendly web tool that allows users to visualize changes in kelp canopy dynamics over time. While the kelp canopy dataset is publicly available [28], the size and file format (netCDF) of the dataset makes it difficult to use for those without data science or coding experience. For example, it does not easily load into commonly used GIS/remote sensing software such as QGIS, ArcGIS, ENVI, or ERDAS IMAGINE. The underlying challenge is the combination of a large geographical extent (Oregon, USA through Baja California Sur, Mexico) and moderate spatial resolution (30 m) that would result in large data files if every single cell, most of them containing open ocean or land (i.e., no kelp canopy), would be represented. Furthermore, the multi-decade length of the dataset, consisting of seasonal means, standard errors, and number of satellite overpasses, makes time series analysis difficult on GIS platforms.

Kelpwatch consists of three elements: (1) a cloud-based backend microservice that makes the data accessible and queryable through an application programming interface (API), (2) a tiling service that serves the classified kelp pixels as a layer to be consumed and displayed by a web map, and (3) a JavaScript-based frontend user interface providing access to the data in any web browser. The frontend was designed to offer simple, intuitive, and informative exploration of the data (Fig 1). Kelpwatch offers users multiple ways to visualize kelp canopy, including animations of dynamics over time and graphs of changes in canopy coverage within

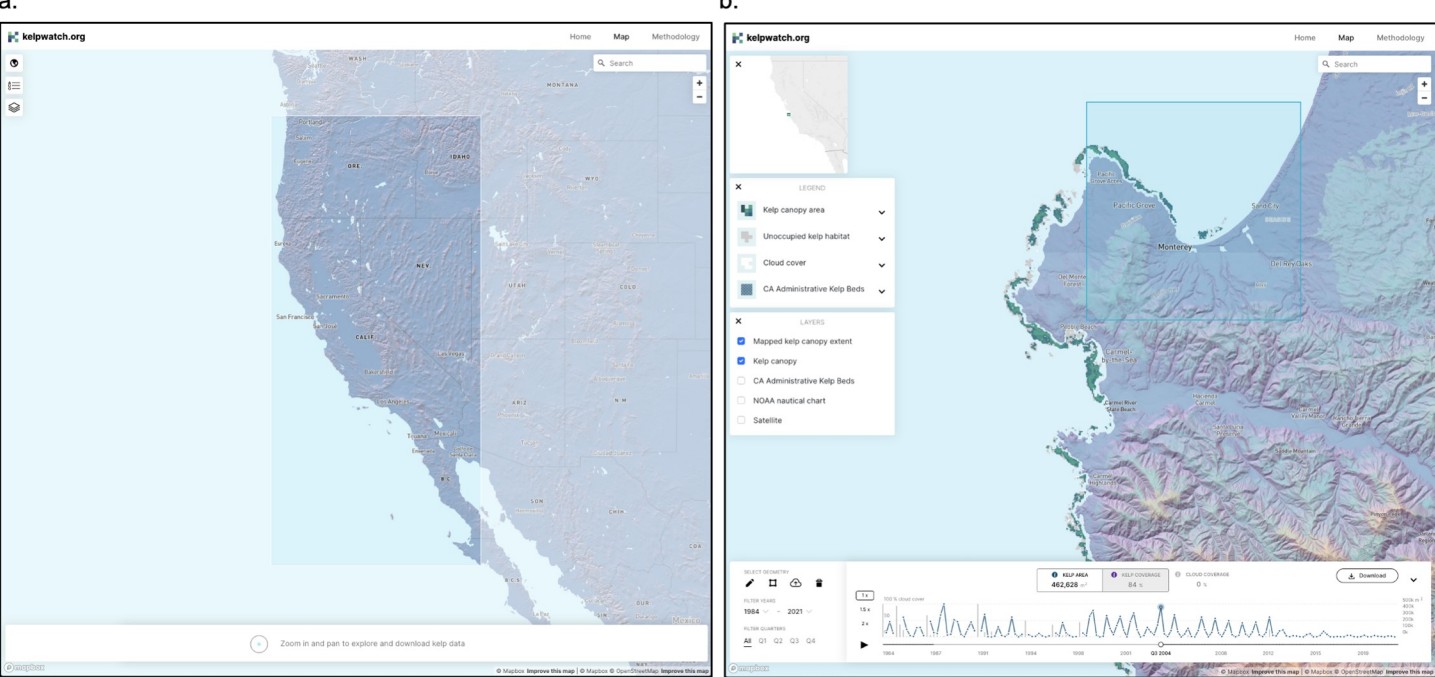

**Fig 1. Kelpwatch web interface.** a.) Visualization of map showing spatial extent of kelp canopy dynamics assessed here. b.) Zoomed in map view of the Monterey Peninsula showing kelp canopy area during the summer of 2004 in shades of blue to green. Unoccupied kelp habitat, or areas where kelp has been observed in the past but is not currently present, is shown in gray and areas obscured by cloud cover for the season (i.e., no data) are shown in white. The blue shaded box shows one of the 10 x 10 km cells used in this study and a seasonal (3-month) time series of kelp area within that cell is shown at the bottom (gray lines represent percent cloud cover of that region through time). Base map from OpenStreetMap and OpenStreetMap Foundation.

a selected area of interest. Importantly, aggregated kelp data within a given geographic and temporal extent can be downloaded as a comma-separated value (csv) file.

## Estimates of kelp canopy area dynamics using Landsat imagery

Estimates of kelp canopy area were determined using a time series of Landsat satellite imagery across the coasts of Oregon and California, USA, and Baja California Norte and Baja California Sur, Mexico. This spatial domain covers regions inhabited by two dominant surface canopy forming kelp species, with bull kelp forming the vast majority of kelp canopies throughout Oregon and northern California, and giant kelp in central and southern California as well as the Baja California Peninsula [1]. Imagery was downloaded as 30 m resolution Collection 1 Level 2 Surface Reflectance products from the United States Geological Survey (https://earthexplorer.usgs.gov/) across four Landsat sensors (Landsat 4 Thematic Mapper, Landsat 5 Thematic Mapper, Landsat 7 Enhanced Thematic Mapper Plus, and Landsat 8 Operational Land Imager) from 1984 to 2021 (S1 Fig and S1 Table). Each Landsat sensor acquires an image every 16 days and an image is provided every eight days when two sensors were operational (1999 to 2011 and 2013 to present). Images were selected if at least part of the coastline was not obscured by cloud cover. Clouds were masked using the pixel quality assurance band included with each image. Since kelp canopy is usually restricted to a narrow swath along the coast (< 1 km) we masked land using a combination of a digital elevation model and a derived intertidal mask. The Advanced Spaceborne Thermal Emission and Reflection Radiometer (ASTER) 30 m digital elevation model (https://www.jspacesystems.or.jp/ersdac/GDEM/E/) was used to identify and mask any pixel with an elevation greater than zero meters. A single

cloud free image for each Landsat path/row acquired at a negative low tide was used to identify intertidal pixels. The Modified Normalized Difference Water Index (MNDWI; [37]) utilizes the shortwave infrared and green spectral bands to derive an index value for each pixel and can separate land exposed during low tide from seawater without interference from floating subtidal kelp canopies. All pixels with a MNDWI index value of less than 0.1 were included in the intertidal mask. After the image classification and processing step described below, pixels adjacent to land with strong, significant negative relationships between the estimated kelp canopy area and tidal height were flagged and plotted on high-resolution satellite imagery in Google Earth to confirm that these pixels were not located in intertidal areas. The pixels were manually removed if they were found to be within the intertidal zone. All tidal height measurements coincided with the time of Landsat image acquisition and were generated from the closest tide station to the center of the image using the Matlab function t_xtide [38].

Once clouds and land were masked, pixels were classified using the binary decision tree classifier described in [18] from band normalized Landsat imagery. The classifier utilizes six spectral bands (blue, green, red, near infrared, and the two shortwave infrared bands) to classify each pixel into four classes (kelp canopy, seawater, cloud, or land; where the cloud and land classes remove pixels not identified in the masks). The classifier was trained on band normalized Landsat imagery where pixels had been grouped using unsupervised k-means clustering (Matlab function kmeans; 15 clusters) and the resulting clusters were then manually assigned one of the four classes. A separate classifier was used for Landsat 8 and Landsat 4/5/7 due to differences in the spectral response functions of the bands. The classifier reduces the number of seawater pixels that are erroneously classified as containing kelp canopy in later processing steps due to sun glint or high seawater turbidity [18]. Multiple endmember spectral mixture analysis (MESMA; [39]) was then used to estimate the proportion of kelp canopy within each 30 m pixel classified as kelp canopy using the blue, green, red, and near infrared spectral bands. A single static kelp canopy endmember was used for all Landsat imagery and 30 seawater endmembers specific to each image were used to account for differences in seawater conditions (e.g., sun glint, phytoplankton bloom, sediment plumes) between locations and image dates [18, 40]. Each pixel was iteratively modeled as the linear combination of the kelp canopy endmember and each seawater endmember, and the resulting kelp canopy fractional cover was determined by selecting the model that minimized the root mean squared error. We estimated the proportional cover of emergent kelp canopy, or the amount of kelp canopy that is directly floating at the surface of the ocean, hereafter referred to as kelp canopy area. Since the MESMA model relies heavily on the near infrared reflectance to estimate the fraction of kelp canopy within each pixel, any portion of the kelp thallus submerged more than a few centimeters below the surface is likely not detected. Fractional kelp canopy within each pixel was converted to kelp canopy area by multiplying the fractional value by the area of the Landsat pixel (900 m$^2$). Canopy area was then provided to Kelpwatch as seasonal (3-month) mean canopy area by calculating the mean kelp canopy area for each pixel for all Landsat images acquired during that quarter [28].

## Regional trends in kelp canopy area

Regional trends in kelp canopy area were assessed by summing all kelp containing pixels within 10 x 10 km cells from the Oregon/Washington border to the southern range limit of canopy forming kelp detected in the time series in Baja California Sur, Mexico. The 10 x 10 km scale was chosen as it provides multiple replicates within each region, but is large enough avoid local-scale processes that may influence kelp canopy dynamics such as sedimentation and recruitment [18, 20]. The corner coordinates of each cell were encoded into GeoJSON

files and uploaded to Kelpwatch. The kelp canopy data was then downloaded from Kelpwatch and cells with less than 500 pixels of potential kelp habitat were excluded from the analysis. The 500 pixel threshold represents that at least 0.5% of the 10 x 10 km cell contains kelp habitat, ensures that each regional-scale cell contains many kelp patches, and that trends are not affected by a single kelp patch within a cell. This pixel threshold resulted in 97.8% of the total kelp canopy area from the complete 30 m scale time series being included within the 10 x 10 km cells. Regional time series from six regions (hereafter referred to as Oregon, Northern California, Central California, Southern California, Baja California Norte, and Baja California Sur; Fig 2A), were produced as a sum of all cells within each region's domain. If greater than 25% of pixels did not have a cloud-free acquisition during a quarter, the entire 10 x 10 km cell was treated as missing data for that quarter. To account for seasonal differences in peak annual canopy area that may result from differences in species phenology or geography [41], the maximum canopy area was determined for each cell for each year from 1984 to 2021. This maximum annual canopy area time series for each 10 x 10 km cell was used for all regional analyses described in this study. Annual canopy area was not determined for a particular year if more than half of that region's cells were missing more than one quarter. The trend of annual kelp canopy area through time in each region, as well as each individual 10 x 10 km cell, was assessed using a generalized least squares regression model (R package nlme; [42]) with an auto-regressive model to account for temporal autocorrelation in the model residuals. As the effects of temporal autocorrelation may change due to species or regional environmental conditions, three auto-regressive processes were applied to each model (zero, first, and second order) and the best model was selected by minimizing the Akaike Information Criterion. To account for differences in available kelp habitat between each 10 x 10 km cell and to express the trend as a percentage increase/decrease, the canopy area time series of each cell was normalized by the maximum canopy area observed between 1984 to 2021.

## Regional marine heatwave response and recovery of kelp canopy

Between 2014 to 2016 the west coast of North America experienced a series of extreme marine heatwaves, with temperature anomalies of 2 to 3°C across the entire California Current [43, 44]. Regional kelp response to and recovery from this heatwave period were assessed by examining the dynamics of kelp canopy within the 10 x 10 km cells relative to the historical annual mean from all years preceding the heatwave period (1984 to 2013). This historical baseline period was characterized by a variety of ocean conditions and includes cool water periods associated with positive North Pacific Gyre Oscillation index values and La Niña events as well as warm water events such as the 1987/1988 and 1997/1998 El Niño events. The response to the marine heatwave period was calculated as the minimum annual kelp canopy area from 2014 to 2016 relative to the historical baseline period. Recovery from the marine heatwave period was calculated as the mean annual kelp canopy area from 2017 to 2021 relative to the historical baseline period. Latitudinal trends in heatwave response and recovery were examined using Pearson correlations, a constant value of 1% was added, and values were log transformed to meet the assumptions of the model.

## Local assessment of kelp canopy area around Monterey Peninsula

The area surrounding the Monterey Peninsula exhibited an overall loss in kelp canopy following the 2014 to 2016 marine heatwave events, with five 10 x 10 km cells showing negative long-term trends, low recovery, and low amounts of recent canopy compared to the historical baseline. To further investigate these patterns at the local-scale (across ~40 km of coastline), kelp canopy area was assessed by summing all kelp containing pixels within 1 x 1 km cells across

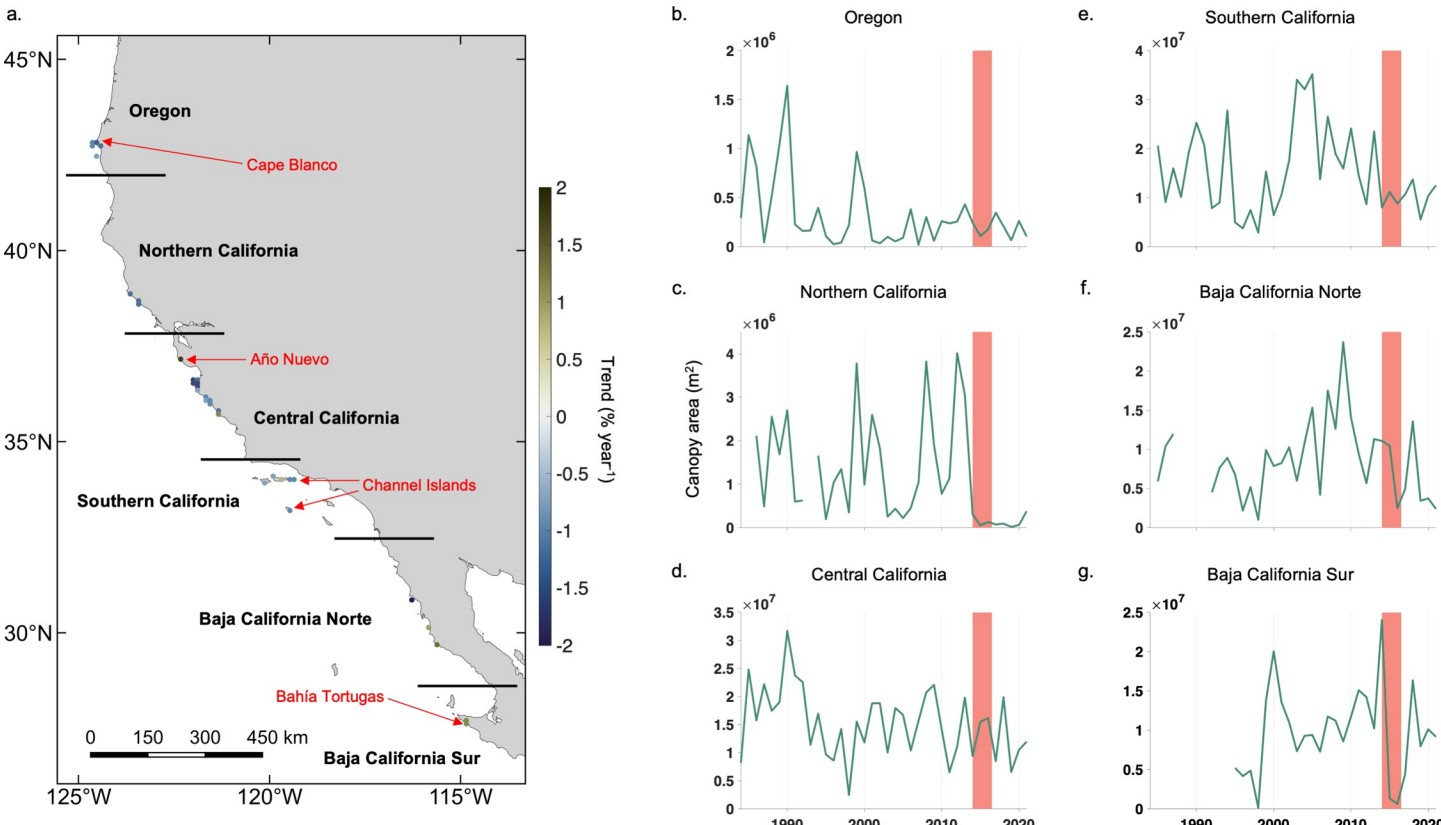

**Fig 2. Study domain and regional time series.** a.) Map of 10 x 10 km cells with significant long-term annual trends across the time series. Regions shown with the horizontal black lines. Place names are shown in red. Annual time series of kelp canopy area from each of the six study regions; b.) Oregon, c.) Northern California, d.) Central California, e.) Southern California, f.) Baja California Norte, g.) Baja California Sur. Vertical red bars show the timespan of the heatwave period (2014 to 2016). Basemap source: Natural Earth.

the affected area. Spatial synchrony of kelp canopy declines dramatically within 200 m [20, 45], so the cell size of 1 km was chosen to avoid spatial autocorrelation processes. Similar to the regional analysis, the corner coordinates of the cells were encoded into GeoJSON files, the data were downloaded from Kelpwatch, and cells with less than 25 pixels of potential kelp habitat were excluded from the analysis. If greater than 25% of pixels did not have a cloud-free acquisition during a quarter, the entire 1 x 1 km cell was treated as missing data for that quarter. Annual maximum kelp canopy area was calculated similarly to the regional analysis and a year with two or more missing quarters was treated as missing data. Mean canopy area from 2014 to 2021 was used to calculate the recent state of the kelp canopy during and since the marine heatwave events relative to the historical mean from 1984 to 2013.

## Results

### Regional trends in kelp canopy dynamics

Kelp canopy area dynamics were assessed for Oregon, Central California, and Southern California regions with either a complete 38-year time series or one missing year. The Northern California and Baja California Norte regions had three and four missing years, respectively (mostly due to the coincidence of cloud cover and a single Landsat sensor early in the time series), while Landsat imagery does not exist for much of Baja California Sur before the mid 1990's resulting in 10 missing years (S1 Table). In Oregon, the vast majority of canopy forming

kelp was observed in the southern portion of the state south of Cape Blanco (Fig 2B). At these large reefs, there were three periods of high kelp canopy across the time series with two events in the mid-1980's to 1990 and one in the late 1990's, which resulted in a significant regional decline of 0.8% per year across the time series (standard error = 0.2%, p = 0.007). The Northern California region displayed multiple periods of high kelp canopy across the time series with peaks in 1999, 2008, and 2012 followed by a period of low canopy after the 2014 to 2016 marine heatwave events (Fig 2C). Due to the high variability in kelp canopy area throughout the time series, there was no significant long-term trend (-0.7% per year, SE = 0.4%, p = 0.123). Central California showed high kelp canopy in the late 1980s and early 1990s with low canopy area during the 1997/1998 El Niño event and from 2019 to 2021 (Fig 2D). High kelp canopy area early in the time series and recent losses resulted in a significant regional decline of 0.6% per year (SE = 0.3%, p = 0.044). Southern California also showed high kelp canopy in the late 1980s and early 1990s and a period of low canopy in the late 1990s (Fig 2E). This period was followed by a rapid recovery after the 1997/1998 El Niño event with peak canopy area from 2003 to 2005. Despite a gradual decline in canopy area since the 2005 peak, there was no significant regional change detected in Southern California across the time series (-0.2% per year, SE = 0.5%, p = 0.711). Baja California Norte also displayed low kelp canopy in the late 1990s followed by a steep increase culminating in peak canopy area in 2009 (Fig 2F). Kelp canopy area has been variable since the peak and no significant regional trend was detected (0.1% per year, SE = 0.4%, p = 0.835). Data for Baja California Sur was missing for the early part of the time series, however there were two distinct kelp canopy peaks in 2000 and 2014 (Fig 2G). There was a steep decline in kelp canopy area during the 2014 to 2016 marine heatwave events with a strong regional recovery in 2018. There was no significant regional trend detected (0.4% per year, SE = 0.5%, p = 0.381).

When examining the 10 x 10 km cells across the entire study domain, 18.6% of the cells displayed a significant trend (p < 0.05; 32 of 172 cells), with 14.5% of the cells showing a negative trend (25 cells) and 4.0% showing a positive trend (7 cells; Table 1). Oregon was the only region that contained a majority of cells with significant trends. The only regions where greater than 10% of cells showed negative long-term trends other than Oregon (55.6%) were Northern California (20.0%) and Central California (35.5%), while Southern California (8.6%) and Baja California Norte (2.9%) had less than 10% of cells with negative trends. Baja California Sur showed no cells with negative long-term trends across the time series. Positive trends were only observed in cells within Central and Southern California, Baja California Norte, and Baja California Sur. Within Central California, cells with negative trends were clustered around and to the south of the Monterey Peninsula while all cells with negative trends in Southern California were found at sites along the offshore Channel Islands (Fig 2A).

## Regional patterns of kelp canopy response and recovery

The response of kelp canopy to the 2014 to 2016 marine heatwave events varied among regions, with kelp canopy area reduced to a minimum annual area of 21.7% of the historical mean (standard deviation 37.8%) across all 10 x 10 km cells (Fig 3). For areas with predominantly bull kelp canopies, Oregon kelp canopy was reduced to 19.0% (17.8%) of the historical regional mean compared to the 2.7% (4.5%) of Northern California canopies remaining during the marine heatwave events. For areas with predominantly giant kelp canopies, Central California was reduced to 56.5% (67.1%) of the historical regional mean, and Southern California and Baja California Norte fared similarly to each other with 14.8% (17.9%) and 24.7% (31.0%), respectively. Kelp canopies in Baja California Sur showed large reductions in kelp canopy during the heatwave period with only 2.2% (5.0%) remaining. The response of kelp canopy area to

**Table 1. Regional trends in kelp canopy.**

| Region | Total Cells | Positive Trend | Negative Trend | No Significant Trend |
|---|---|---|---|---|
| Oregon | 9 | 0 (0%) | 5 (55.6%) | 4 (44.4%) |
| Northern California | 15 | 0 (0%) | 3 (20.0%) | 12 (80.0%) |
| Central California | 31 | 2 (6.5%) | 11(35.5%) | 18 (58.0%) |
| Southern California | 58 | 1 (1.7%) | 5 (8.6%) | 52 (89.7%) |
| Baja California Norte | 35 | 2 (5.7%) | 1 (2.9%) | 32 (91.4%) |
| Baja California Sur | 24 | 2 (8.3%) | 0 (0%) | 22 (91.7%) |

Total number of 10 x 10 km cells and the number (percent) of cells with positive, negative, no significant long-term annual trends for each region.

the marine heatwave events was significantly related to latitude for both giant kelp and bull kelp dominated regions. On the 10 x 10 km scale there was decreased log kelp canopy area with decreased latitude from Baja California Sur to Central California (giant kelp dominated; $r = 0.562$, $p < 0.001$) and from Northern California to Oregon (bull kelp dominated; $r = 0.672$,

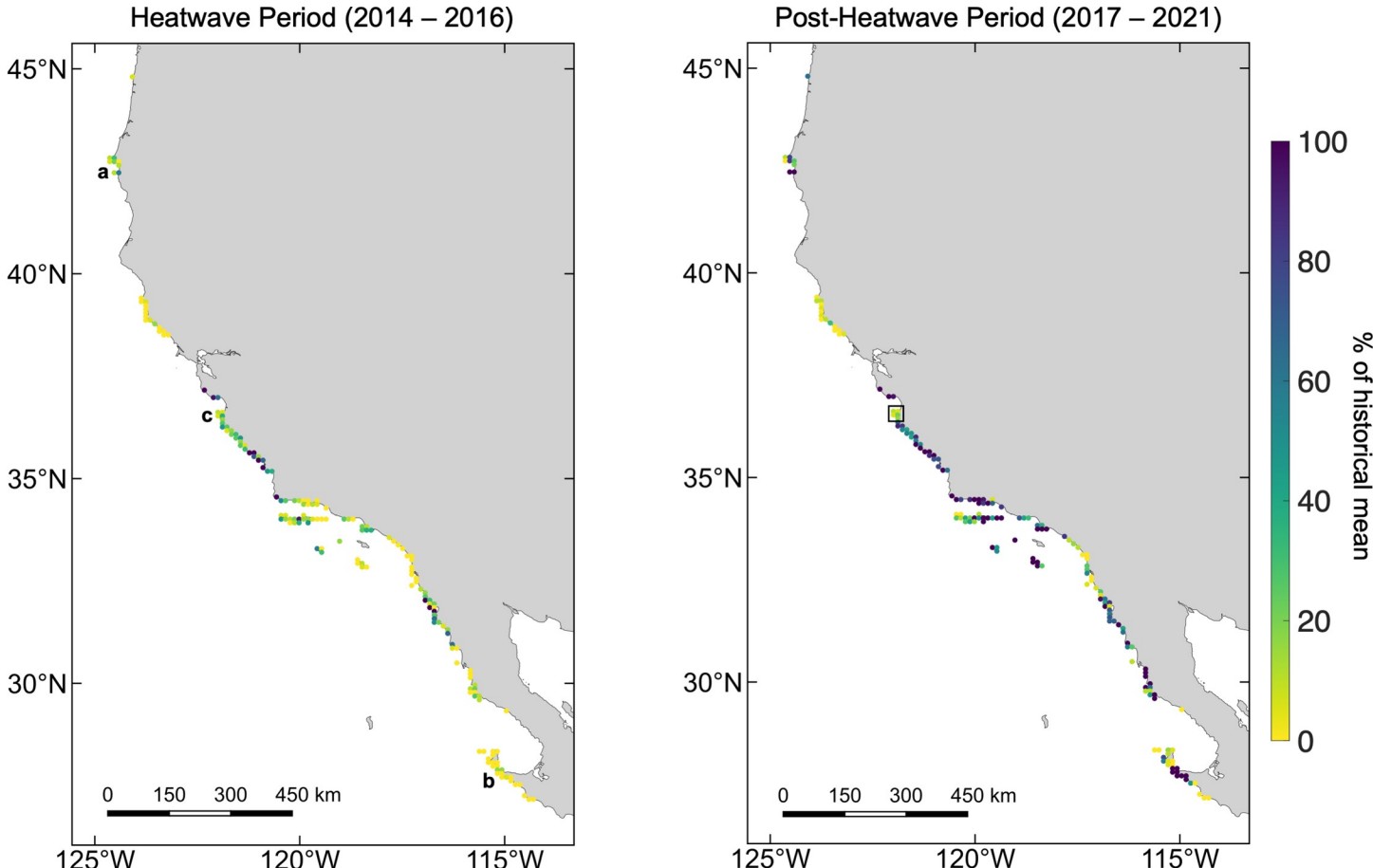

**Fig 3. Response to and recovery from the marine heatwave events.** Maps of kelp canopy response to the 2014 to 2016 marine heatwave events (minimum annual kelp canopy area remaining during the heatwave period as a percentage of the historical mean) and recovery (mean annual kelp canopy area after the heatwave period as a percentage of the historical mean) within the 10 x 10 km cells for the study domain. The letters correspond to the locations shown in Fig 5. The black box corresponds to the area shown in Fig 6. Basemap source: Natural Earth.

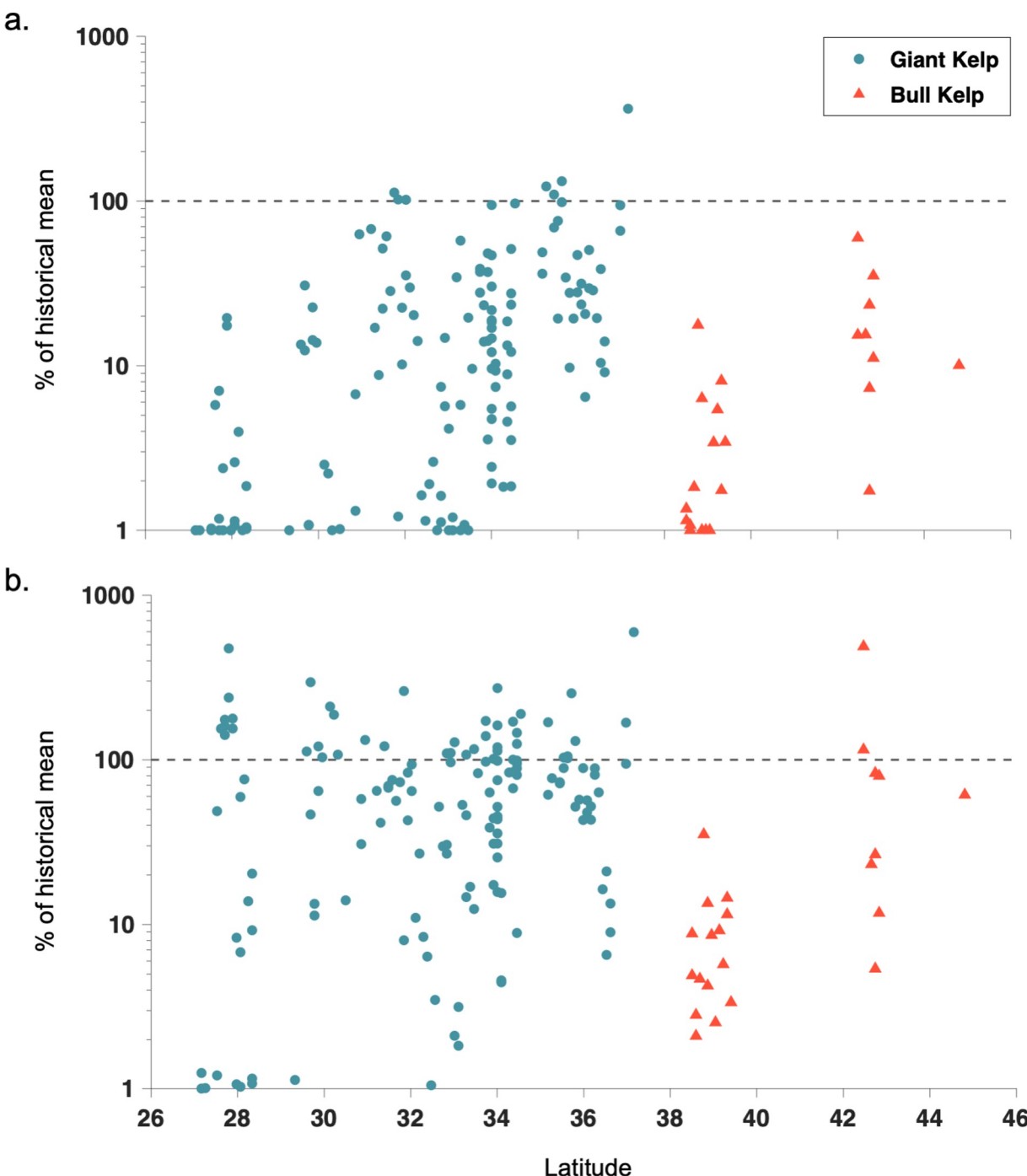

**Fig 4. Latitudinal trends in response and recovery.** Scatter plots of kelp canopy a.) response to and b.) recovery from the 2014 to 2016 marine heatwave events compared to the historical mean (1984 to 2013) for each 10 x 10 km cell. Areas dominated by giant kelp canopy are shown as blue circles and areas dominated by bull kelp canopy are shown as red triangles. Black horizontal dashed lines show the historical mean. Note that the y-axes are on a log scale.

p < 0.001) although there was local-scale variability among the 10 x 10 km cells (Fig 4A; for untransformed data see S2A Fig).

Recovery from the 2014 to 2016 marine heatwave events varied both among and within regions, with a mean recovery of 73.0% (standard deviation 85.8%) across all 10 x 10 km cells (Fig 3). Recovery within each region for areas dominated by giant kelp led to a weaker significant relationship between log recovery and latitude (r = 0.263, p = 0.001; Fig 4B; for untransformed data see S2B Fig). However, the relationship was similar to the heatwave response for bull kelp dominated areas (r = 0.681, p < 0.001). While kelp canopy recovery in Northern California remained low after the marine heatwave events with only a 7.8% (standard deviation 8.3%) recovery compared to the historical mean, kelp canopies in Oregon were highly variable, with a mean regional recovery of 98.3% (150.5%). These ranged from less than 10% recovery at the offshore areas of Orford Reef to greater than 480% recovery at Rogue Reef (Fig 5A). Recovery significantly greater than 100% (e.g., Rogue Reef), may indicate that the 2014 to 2016 marine heatwave was not a dominant driver of canopy area dynamics in this sub-region. In the giant kelp dominated regions, recovery was strong along most of the Central California coast, apart from kelp canopies around and to the south of the Monterey Peninsula, with a mean regional recovery of 96.8% (107.9%). Recovery across Southern California was variable with a mean regional recovery of 68.9% (54.5%) with areas of low recovery occurring in the western half of the Northern Channel Islands and the southern portion of the region. Similarly, recovery in Baja California Norte was also variable with a mean regional recovery of 75.7% (71.5%). The Baja California Sur region showed a mean regional recovery of 79.4% (113.4%) with strong local variability, for example areas around Bahía Tortugas showed high recovery while areas to the immediate north and south displayed little recovery (Fig 5B). Overall, 22.2% of the 10 x 10 km cells in Oregon recovered to or above the historical mean while 0% of Northern California cells fully recovered. Central and Southern California and Baja California Norte and Sur all had a similar proportion of cells recover at levels greater than their historical mean, 29.0%, 25.9%, 28.6%, and 33.3% respectively.

## Local patterns of kelp canopy around Monterey Peninsula

The cluster of five 10 x 10 km cells surrounding and to the south of the Monterey Peninsula showed some of the strongest declines across the time series (Figs 2A and 5C), low recovery from the marine heatwaves despite relatively high recovery across most of the Central California region (Fig 3), and low overall kelp canopy during and after the heatwave events compared to the historical mean (Fig 6). The seasonal time series of kelp canopy area for the Monterey Peninsula displayed steep declines during the 2014 to 2016 marine heatwave events and continued to decline post-heatwave (Fig 6A). The period of decline (2014 to 2021) represents the longest consecutive period of low canopy kelp cover around and to the south of the Monterey Peninsula in the entire time series, with canopy area during this period representing 17.4% of the mean canopy area from 1984 to 2013. When examined at the local-scale (1 x 1 km cells), 91.1% of cells showed less than 50% of their historical mean during the period from 2014 to 2021 (Fig 6B). Furthermore, 46.7% of the cells were less than 10% of their historical mean, while only 6.7% of cells were at or above their historical mean.

## Discussion

### Long-term trends in kelp canopy across regions

Significant long-term trends in kelp canopy area were observed at the 10 x 10 km scale and across entire regions. Kelp canopy in Oregon was characterized by three short periods of high canopy area early and in the middle of the time series, followed by a longer period of moderate kelp canopy starting the mid-2000's, resulting in a significant, overall regional decline from 1984 to 2021 (Fig 2B). Total kelp canopy area in Oregon was dominated by the dynamics on

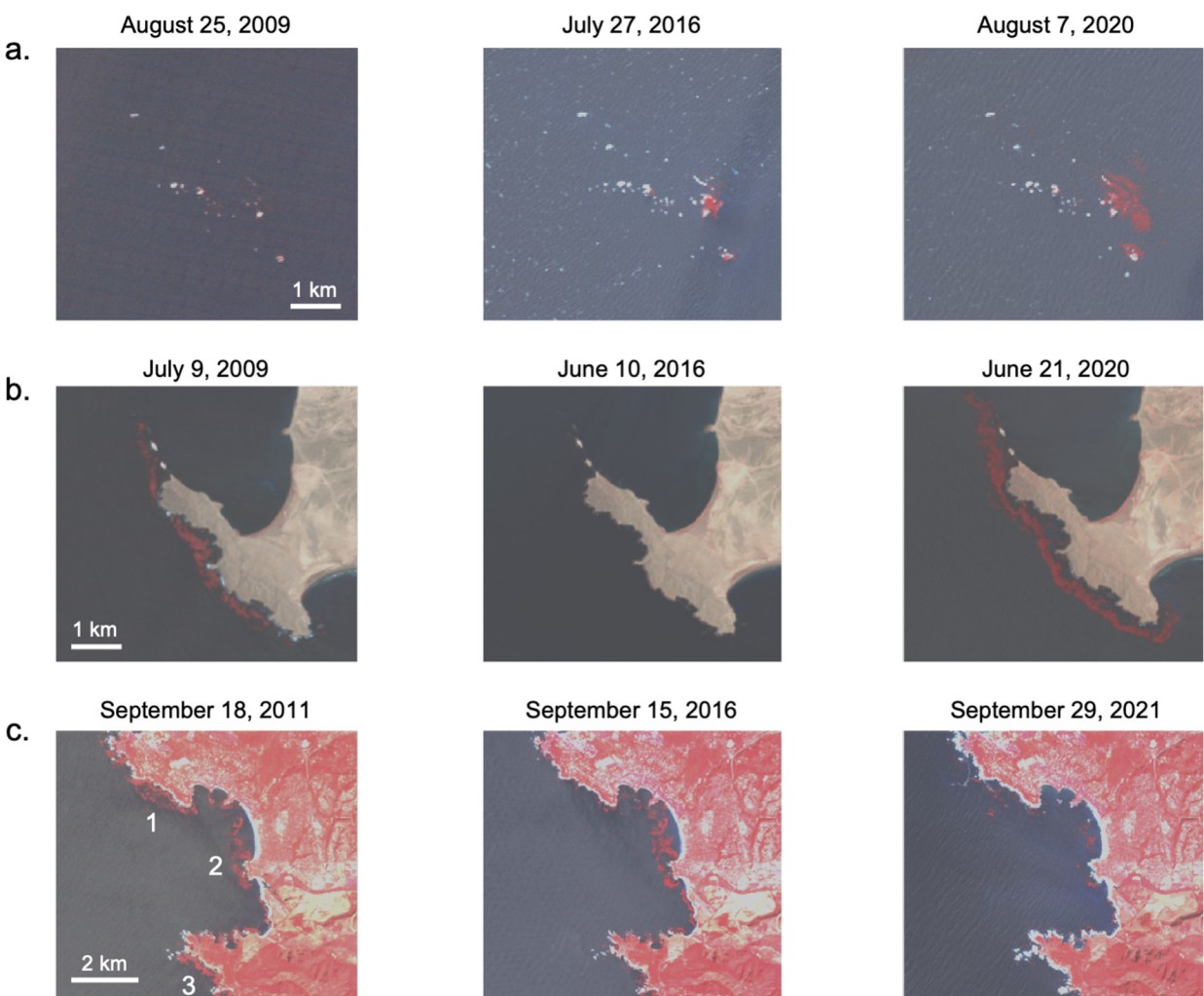

**Fig 5. False color Landsat images showing change in kelp canopy.** a.) Rogue Reef, Oregon, b.) Bahía Tortugas, Baja California Sur, and c.) Monterey Peninsula and Carmel Bay, California before, during, and after the marine heatwave events. Locations shown in Fig 3. Kelp canopy has high reflectance in the near infrared which has been colored red in these images for visualization purposes. The numbers in bottom left image represent 1. Pescadero Point, 2. Carmel Point, and 3. Point Lobos. Landsat imagery from USGS.

Orford, Blanco, and MacKenzie reefs (Fig 2A), a large rocky reef complex southwest of Cape Blanco totaling ~50 km$^2$ with depths ranging from 10 to 25 m [46]. Five of the nine 10 x 10 km cells in the Oregon region declined over the study period, the only region where a majority of cells showed significant long-term trends (Table 1). While there are a paucity of studies examining subtidal kelp dynamics and environmental drivers in Oregon, Hamilton and others [14] used Landsat imagery to assess five of the region's largest kelp forests from 1984 to 2018. Two of the five forests examined displayed long-term declines, including Orford Reef [14]. Kelp forests in Oregon have also been periodically assessed via aerial imagery by the Oregon Department of Fish and Wildlife with imagery collected in 1990, 1996 to 1999, and 2010 [46–50]. A change in the image collection methodology from color-infrared photography (1990's) to digital multispectral imagery (2010) made intercomparison difficult as photographs from the 1990's were delineated by hand while individual 1-meter pixels were classified as kelp canopy from the multispectral imagery [50]. Without robust calibration between sensors through time, as done with the Landsat sensors [18], long-term trend analysis is impossible.

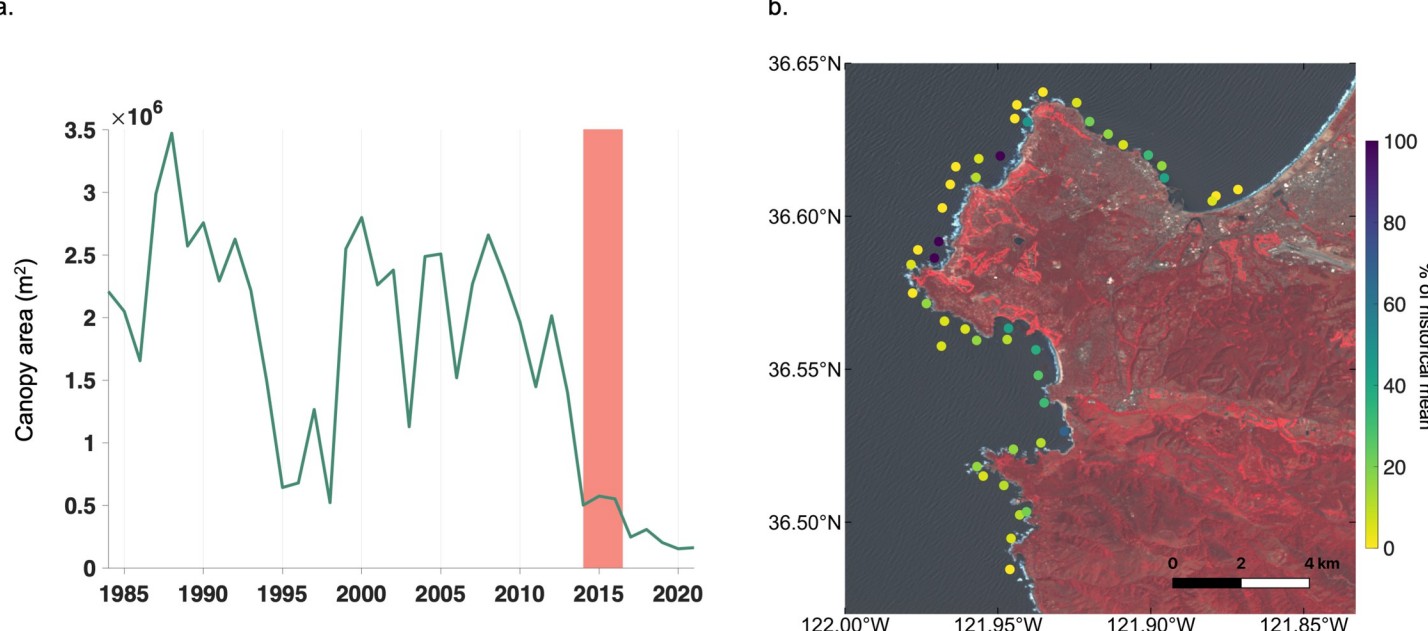

**Fig 6. Kelp dynamics around the Monterey Peninsula.** a.) Time series of maximum annual kelp canopy area for the Monterey Peninsula. Vertical red bar shows the timespan of the heatwave period (2014 to 2016). b.) Mean annual canopy area from 2014 to 2021 in 1 x 1 km cells as a percentage of the historical mean annual canopy area from 1984 to 2013. Values of 100% represent kelp canopy areas greater than or equal to the historical mean. Landsat imagery from USGS.

Kelp forests off the coast of Northern California have experienced historic lows from 2014 to 2021 [11, 13, 24], however, while 20% of the 10 x 10 km cells showed a negative trend, a significant regional decline was not observed in the full time series. Kelp canopy was highly dynamic across the region, with three periods of high multiyear kelp canopy area during the late 1980's to early 1990's, late 1990's to early 2000's, and late 2000's to early 2010's, with the two highest years occurring in 2008 and 2012 (Fig 2C). Despite the recent period of low or absent kelp canopy across the region from 2014 to 2021, high levels of canopy present in the years immediately preceding the marine heatwave (2008, 2012 to 2013) demonstrates the interannual oscillatory nature of the kelp canopy in this region. The Central California region did display a significant regional decline in canopy area across the time series (Fig 2D), driven by a large decrease in canopy area around the Monterey Peninsula since 2014 (Fig 3B; see *Local Declines Around Monterey Peninsula*). High kelp canopy area early in the time series resulting from both high canopy density and large canopy extent likely contributed to this negative trend. The Central California region represents a transition zone between the two major canopy forming kelp species but is mostly dominated by giant kelp [1]. While over 35% of 10 x 10 km cells displayed negative trends (Table 1), Central California also possessed two cells with positive trends, including the northernmost cell near Point Año Nuevo, which showed increasing post-heatwave canopy area.

Southern California and Baja California Norte both showed a low percentage of cells with significant trends (~10%), with every Southern California cell with significant trends located on the offshore Channel Islands (Fig 2A). Both regions suffered major reductions in plant and stipe density and canopy area during the 1997/1998 El Niño event, reaching regional minimums in 1998 (Fig 2E and 2F; [51, 52]). However, both regions responded positively to the 1999/2000 La Niña event and canopy area continued to increase regionally with positive North Pacific Gyre Oscillation index values and the associated elevated seawater nutrients [6, 53]

with regional maximums in 2005 for Southern California and 2009 for Baja California Norte (Fig 2F and 2G). Baja California Sur had the shortest regional time series due to a lack of available imagery at the beginning of the time series (S1 Table). While no regional trend was detected, the two positive long-term trends from individual 10 x 10 km cells should be treated with skepticism since missing data occurred during a period when canopy area was relatively high across regions.

The relationship between regional kelp canopy dynamics and decadal marine climate oscillations [6, 40, 54] produce multiyear periods of high (or low) kelp canopy that make the identification of long-term trends difficult [18]. This interannual oscillatory nature of regional kelp canopy dynamics is apparent in the regional time series and may have resulted in greater then 80% of 10 x 10 km cells showing no significant long term trend (Fig 2). A recent analysis has shown that the synchrony of giant kelp canopy is highly coherent with the North Pacific Gyre Oscillation on long time scales (4 to 10 years; [55]), meaning that sites within regions tend to increase and decrease similarly according to the fluctuations of the large-scale ocean climate. Since regular oscillatory patterns make the detection of long-term trends difficult [18, 56], perhaps the most beneficial use for these data is to investigate the spatial heterogeneity of the response in kelp canopy to major climate events, such as the 2014 to 2016 marine heatwaves. Here, this type of analysis uncovered sub-regional (and potentially local-scale) variability in kelp canopy and could allow researchers to hone in on areas showing disparate patterns and elucidate underlying drivers.

## Kelp canopy response to and recovery from the 2014 to 2016 marine heatwave events

During the summer of 2014, an unprecedented warm water temperature event spread across the Northeastern Pacific leading to negative impacts across both nearshore and pelagic ecosystems [43, 57, 58]. This marine heatwave, known as 'The Blob', was closely followed by a strong El Niño event in 2015 to 2016, contributing to an extended period of anonymously high ocean temperatures, low seawater nutrients, and low productivity throughout the region [59]. Notably, there were historic and widespread declines in kelp forest ecosystems associated with these events, both in Northern California [13] and Baja California [60], although ecosystem response was not constant across regions [61]. We found that kelp canopies across all regions declined in response to the marine heatwave period, but that these declines were significantly related to latitude. Cavanaugh and others [10] found that the resistance/response of giant kelp canopy across Southern and Baja California was associated with an absolute temperature threshold (23˚C) and not a relative temperature anomaly. Since ocean temperature generally decreases in the California Current with increasing latitude [62], it is perhaps not surprising that the response of giant kelp canopy to the marine heatwave events was more negative in southern regions given that giant kelp responds strongly to temperatures over an absolute threshold. Interestingly, regions that are primarily composed of bull kelp also exhibited a similar latitudinal response separate from the one displayed by regions dominated by giant kelp (Fig 4A). This implies that each species may possess specific temperature thresholds for growth and mortality and in fact, recent laboratory experiments show that bull kelp blades maximize elongation rate at 11.9˚C with precipitous declines at temperatures above 16˚C [63].

While previous marine heatwave events have resulted in short-term declines in kelp abundance across regions, the recovery of the kelp canopy following the heatwave can be spatially variable and often occur at smaller spatial scales (meters to kilometers; [51]). While a significant positive relationship between canopy recovery and latitude was found for giant kelp, the relationship was more variable when compared to its response to the heatwave period, with

examples of high recovery in all four giant kelp dominated regions (Fig 4B). This is a similar result to previous studies that found no clear relationship between large-scale high ocean temperatures and heatwave variables to kelp recovery [10]. One striking example of kelp canopy recovery is the kelp forest that surrounds Bahía Tortugas in Baja California Sur near the southern range limit for giant kelp in the Northern Hemisphere (Fig 5B). This kelp forest displayed high canopy area before the heatwave, a complete loss of canopy during the heatwave, and a complete recovery to greater canopy area in the years following. However, this kelp forest is surrounded by 10 x 10 km cells that displayed little recovery during the five years post-heatwave events (Fig 3), implying a driver acting over a smaller spatial scale than a latitudinal temperature gradient. The coast of Baja California has a varied geometry leading to distinct sub-regional upwelling zones that are oriented parallel to the dominant wind direction [64]. The three upwelling zones located within the Baja California study domain exist at 31.5˚N, 29˚N, and 27˚N and all correspond to cells with high recovery. The sub-regional nature of coastal upwelling, delivering cool, nutrient-rich seawater to the nearshore, may be vital to kelp forest recovery after heatwave events and future studies comparing kelp dynamics to localized upwelling are needed. Regions dominated by bull kelp showed a significant relationship between recovery and latitude, driven by little recovery in Northern California and varied recovery in Oregon (Fig 4B). While signs of kelp canopy recovery in Northern California did not begin until 2021, some sites in Oregon displayed increases in canopy area throughout and after the heatwave events. An example of this is Rogue Reef, where little canopy was present prior to the heatwave, small increases occurred during the heatwave, and a large canopy formed post-heatwave (Fig 5A). This incredible level of recovery versus the historical mean canopy (~480%) represents one of the few areas with increasing kelp canopy during the marine heatwave events. As there are fewer subtidal monitoring programs in Oregon compared to California [65], more work is needed to understand the spatial drivers of kelp forest dynamics across Oregon.

## Local declines around Monterey Peninsula

While the Central California region exhibited relatively high levels of recovery to the marine heatwave events, the five 10 x 10 km cells surrounding the Monterey Peninsula showed less than 20% recovery compared to the historical mean (Fig 3). Prior to the heatwave events, kelp canopy area around the Monterey Peninsula was seasonally dynamic, with large winter waves removing whole plants and/or canopy each year leading to a reduction in kelp abundance [6, 66]. However, kelp canopies in this subregion were persistently high on annual time scales, with decreases during the heatwave events of 2014 to 2016 and with further reductions post-2016 (Fig 6A). This cluster of cells with low sustained recovery warranted a local-scale analysis made possible by altering the domain of the spatial input polygons uploaded to Kelpwatch. During the post-heatwave years (2014 to 2021) the vast majority of 1 x 1 km cells showed less than 50% of their mean historical canopy area (1984 to 2013) with only a few local-scale examples of high recovery (Fig 6B). An examination of the Landsat imagery used to generate the kelp canopy data for Kelpwatch further illustrates these declines. The kelp forests near Pescadero Point (Fig 5c1), Carmel Point (Fig 5c2), and Point Lobos (Fig 5c3) can be clearly seen as the offshore red patches in the false color imagery during September 2011. By September 2016, the Pescadero Point kelp forest canopy (1) had nearly disappeared, Carmel Point (2) was similar in area to 2011, and Point Lobos (3) had been reduced to a few patches. By September 2021, the Pescadero Point kelp forest (1) was showing some patchy recovery, Carmel Point (2) had been reduced to patches and Point Lobos (3) had nearly disappeared. These spatial patterns exposed by Kelpwatch support a recent field-based analysis examining the role of sea otters, an

important predator of herbivorous sea urchins. Smith and others [15] found that the spatial pattern of sea otter foraging was associated with the distribution of energetically profitable urchins, that is, restricted to areas that maintained high kelp densities and well-fed sea urchins. This resulted in a patchy mosaic of kelp forest stands interspersed with sea urchin barrens, possibly enhancing the resistance of existing stands but not directly contributing to the resilience of areas without kelp [15]. While this explains the spatial patchiness and lack of recovery of kelp canopy around the Monterey Peninsula, it does not explain why this subregion showed less recovery than other areas in Central California. The sustained decline in kelp canopy around the Monterey Peninsula detected using the Kelpwatch tool represents the longest period of low canopy cover for this area over the length of the Landsat time series, suggesting that more research and monitoring attention should be directed at this location. Understanding differences in environmental conditions and trophic interactions around the Monterey Peninsula and nearby locations that have exhibited high kelp canopy recovery may shed light on important drivers that are best assessed by instrumented moorings and diver-based survey methods. This case study demonstrates how a decision-support tool like Kelpwatch can be used to make complex data actionable for managers, restoration practitioners, and researchers, and promote data-driven resource management.

## Conclusions

Over the past decade, there has been an increased focus on the long-term declines of kelp forests both regionally and globally, usually in the context of warming ocean conditions, competition with other reef space holders, and increases in herbivore abundance [67–70]. While kelp forests in many regions have undoubtedly experienced severe and unprecedented declines in recent years [11, 71], time series of kelp dynamics are often limited by short durations or punctuated field campaigns. These time series limitations can obscure the true nature of kelp forest change especially given the rapid dynamics of kelp. For example, a recently observed decline may be related to a decadal marine climate oscillation and similar periods of low kelp abundance may have occurred before the time series was initiated or were missed due to logistical or funding constraints. Therefore, it is essential to produce a long, continuous, and calibrated time series in order to put recent declines in the context of long-term dynamics. While Landsat observations can be limited by cloud cover and can only detect fluctuations in surface canopy, the uninterrupted satellite continuity (1984 to present), rapid repeat frequency (16 days from 1984 to 1998; 8 days from 1999 to present), and large spatial domain (global) offer an unparalleled opportunity to track kelp forest dynamics [18, 72]. The assessment of continuous kelp dynamics allows for the observation of decadal cycles in canopy cover that often result from changes in regional nutrient regimes (e.g., the North Pacific Gyre Oscillation; [6, 40, 53]) or sudden regional-scale crashes and recoveries in canopy cover resulting from El Niño and La Niña events, respectively [21]. Recently, an ensemble of climate models was used to determine the appropriate time series length needed to distinguish a climate change precipitated trend from natural variability for several biogeochemically relevant marine variables and found that time series of at least 40 years in length are necessary to define the natural variability of biotic variables (phytoplankton chlorophyll concentration and production dynamics; [56]). With close to 40 years of observations as of the time of this analysis, the Landsat-derived data available on Kelpwatch are beginning to approach the length necessary to observe changes in kelp canopy across decadal cycles and detect long-term trends.

Tools like Kelpwatch make earth observation data actionable and will help scientists and managers identify areas to focus research and monitoring efforts to understand how kelp forests respond to marine heatwaves and other pressures, and to place these dynamics in

historical context to inform strategic management interventions. The analyses in this study are descriptive in nature and further work by researchers and kelp forest managers is required to identify the drivers of kelp canopy response to and recovery from disturbance events such as the 2014 to 2016 heatwave period. The near coincident occurrence of high ocean temperatures, reduced seawater nutrient concentrations, and increased density of herbivorous sea urchins during the heatwave period necessitate additional spatial data streams to identify driver timing and strength [11]. Additionally, the recovery of kelp populations is subject to additional demographic (e.g., spore supply; [73]) and ecological processes such as hysteresis [74] that may cloud the recovery of population dynamics if only abiotic drivers are considered [10]. All the analyses in this study were completed using data downloaded directly from the Kelpwatch platform and analyzed in commonly used geospatial or statistical programs for transparency and reproducibility. Having a calibrated, open-access, and continuous time series of kelp canopy dynamics puts the ability to examine near real-time observations of kelp canopy and spot problem areas for kelp loss in the hands of scientists and managers.

## Supporting information

**S1 Fig. Study domain.** Landsat tile composite of the study domain labeled with its corresponding path/row number. Landsat imagery from USGS.
(TIFF)

**S2 Fig. Latitudinal trends in response and recovery.** Scatter plots of kelp canopy a.) response to and b.) recovery from the 2014–2016 marine heatwave events compared to the historical mean (1984 to 2013) for each 10 x 10 km cell. Areas dominated by giant kelp canopy are shown as blue circles and areas dominated by bull kelp canopy are shown as red triangles. Black horizontal dashed lines show the historical mean.
(TIFF)

**S1 Table. Number of Landsat images by year.** The number of Landsat images used for each year for each path/row.
(TIFF)

## Acknowledgments

We thank Steve Lonhart, Kristen Elsmore, and Dana Morton for reviews of the manuscript.

## Author Contributions

**Conceptualization:** Tom W. Bell, Kyle C. Cavanaugh, Vienna R. Saccomanno, Norah Eddy, Falk Schuetzenmeister, Nathaniel Rindlaub, Mary Gleason.

**Data curation:** Tom W. Bell, Kyle C. Cavanaugh,  Katherine C. Cavanaugh, Henry F. Houskeeper, Falk Schuetzenmeister, Nathaniel Rindlaub.

**Formal analysis:** Tom W. Bell, Kyle C. Cavanaugh.

**Funding acquisition:** Tom W. Bell, Kyle C. Cavanaugh, Vienna R. Saccomanno, Norah Eddy, Mary Gleason.

**Investigation:** Tom W. Bell, Vienna R. Saccomanno, Katherine C. Cavanaugh.

**Methodology:** Tom W. Bell, Kyle C. Cavanaugh, Katherine C. Cavanaugh, Henry F. Houskeeper, Falk Schuetzenmeister, Nathaniel Rindlaub.

**Project administration:** Tom W. Bell, Norah Eddy, Falk Schuetzenmeister, Nathaniel Rindlaub, Mary Gleason.

**Resources:** Mary Gleason.

**Software:** Falk Schuetzenmeister, Nathaniel Rindlaub.

**Supervision:** Kyle C. Cavanaugh, Vienna R. Saccomanno, Norah Eddy, Falk Schuetzenmeister, Nathaniel Rindlaub, Mary Gleason.

**Validation:** Tom W. Bell, Katherine C. Cavanaugh, Henry F. Houskeeper.

**Visualization:** Tom W. Bell, Falk Schuetzenmeister, Nathaniel Rindlaub.

**Writing – original draft:** Tom W. Bell, Kyle C. Cavanaugh.

**Writing – review & editing:** Tom W. Bell, Kyle C. Cavanaugh, Vienna R. Saccomanno, Katherine C. Cavanaugh, Henry F. Houskeeper, Norah Eddy, Falk Schuetzenmeister, Nathaniel Rindlaub, Mary Gleason.

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
