## [Decision Letter · Decision Letter 0]

3 Aug 2022

PONE-D-22-18739Kelpwatch: A new visualization and analysis tool to explore kelp canopy dynamics reveals variable resistance and resilience to marine heat wavesPLOS ONE

Dear Dr. Bell,

Thank you for submitting your manuscript to PLOS ONE. After careful consideration, we feel that it has merit but does not fully meet PLOS ONE’s publication criteria as it currently stands. Therefore, we invite you to submit a revised version of the manuscript that addresses the points raised during the review process.

We look forward to receiving your revised manuscript.

Kind regards,

Alejandro Pérez-Matus

Academic Editor

PLOS ONE

Journal Requirements:

3. We note that Figures 1, 7, S1 and S2 in your submission contain [map/satellite] images which may be copyrighted. All PLOS content is published under the Creative Commons Attribution License (CC BY 4.0), which means that the manuscript, images, and Supporting Information files will be freely available online, and any third party is permitted to access, download, copy, distribute, and use these materials in any way, even commercially, with proper attribution. For these reasons, we cannot publish previously copyrighted maps or satellite images created using proprietary data, such as Google software (Google Maps, Street View, and Earth). For more information, see our copyright guidelines: http://journals.plos.org/plosone/s/licenses-and-copyright.

a. You may seek permission from the original copyright holder of Figures 1, 7, S1 and S2 to publish the content specifically under the CC BY 4.0 license.  

Additional Editor Comments:

I have received two reviews from experts on the subject, and although both consider the work very positive and potentially find it to be a useful tool to inform the management of this large-scale ecosystem, they consider that there should be improvements on some interpretations and presentation of results. One of the most important considerations is the mix that gives rise to misconceptions of the stability of these ecosystems and their dimensions such as resistance and resilience. I invite the authors to improve these interpretations by following some constructive recommendations from the reviewers.

Reviewers' comments:

Reviewer's Responses to Questions

**Comments to the Author**

1. Is the manuscript technically sound, and do the data support the conclusions?

Reviewer #1: Partly

Reviewer #2: Yes

2. Has the statistical analysis been performed appropriately and rigorously? 

Reviewer #1: No

Reviewer #2: Yes

3. Have the authors made all data underlying the findings in their manuscript fully available?

Reviewer #1: Yes

Reviewer #2: Yes

4. Is the manuscript presented in an intelligible fashion and written in standard English?

Reviewer #1: Yes

Reviewer #2: Yes

5. Review Comments to the Author

Reviewer #1: This manuscript reports on the dynamics of kelp canopy over a long period (>35years) from satellite data analysis along the Pacific coast of North America. The analysis is based on a new web-based tool to visualize kelp canopy from the satellite imagery database. The tool seems very interesting as it allows visualizing time series of kelp canopy at local scale and downloading parts or all of the data base for personal purposes. The manuscript illustrates the possible use of this tool by presenting an analysis of small-scale and large-scale dynamics of kelp canopy associated with a period of strong heatwaves.

Several flaws are to be considered, however, as they cause important confusions and possibly distortions of the actual dynamics the authors want to describe.

The first concerns the extremely limited explanation of the method to detect canopy cover from satellite images. I understand this method is explained in other papers, but as the purpose of this paper is to present the Kelpwatch database, a brief explanation of the essential aspects of kelp canopy detection and the classification of pixels would be really helpful. On the contrary, the description of the treatment of cloudy images and coastal areas are thoroughly detailed (and long). Also, I did not understand the procedure of first classifying pixels based on multispectral images and then performing MESMA. If MESMA identifies kelp canopy, then how/why were pixels previously classified as bearing or not kelp canopy? Some details are mentioned but need explanation: for example, p7, “Collection 1 Level 2 Surface Reflectance” is quite obscure for the non-specialist. A description (and/or a mention of its properties) would be useful to understand the approach. A justification for the choice of leaving cells with less than 500 pixels of kelp habitat is also needed, together with an estimate of the area that this represents (or at least the % of the total area of a cell). Overall, the section describing the methods implemented to generate the database should be meticulously revisited.

The second flaw concerns the concepts of resistance and resilience. Here, resistance is considered as the ratio of canopy extension during the 2014-2016 period over the prior extension, and resilience was calculated as the fraction of initial cover that was recovered after that period. These are relatively poor indicators of the concepts of resistance and resilience, if considered alone. The authors use a 5-years average to define a baseline for the initial cover. This eventually introduces strong bias: if cover increased from 1 to 10 during that period, then the loss of cover calculated from the average (let’s say 5) would be half of what was actually lost during the perturbation. Similar reasoning for a 5-years period that would have experienced a decline from 10 to 1 with an average of 5, but with opposite effect on the estimation of canopy resistence. Similar effects can also be found in the way resilience was calculated (also using a 5-years average posterior to 2014-2016). Resilience is generally defined in terms of recovery, 100% resilience usually being the maximum value because it refers to the return to a state prior to a perturbation. Here, values as high as 1400% recovery makes very little sense under the concept of resilience. I strongly suggest the authors to revisit how resistance and resilience are calculated in ecology and population dynamics. Several reviews are available (I suggest Capedevilla et al. 2020, TREE, as one of the most recent) to define better estimates. By looking at the strong variance in kelp cover, which often does not follow any trend, one way of looking at resistance and resilience would be using a variance baseline, instead of an average. If kelp canopy varies x% over normal years, then resistance should be how much the variance during 2014-2016 exceeds the previous 5-y. And resilience could be determined as the capacity to reach the confidence interval of that period. Eventually, the timing for recovery would be an interesting information. Finally, because surface canopy area nearly reaches 0 each winter, years of high canopy area likely correspond to high growth rates and/or biomass accumulation during spring and summer of that year. Therefore, recovery as it is defined here only indicates the capacity of the population to produce biomass. As it is the case for Orford Reef (and its 1400% recovery), the production of biomass during a specific year may be completely independent of the biomass during the previous year if individuals are perennial or the population is persistent through an important gametophyte bank. This specific point should be discussed, to avoid misinterpretations about population dynamics.

Another flaw is the focus on so called trends. While long term trends in canopy extension can be an interesting indicator about population persistence, growth or decline, other trends examined in this study seem poorly supported. The latitudinal trend is one of them, as temperature (average, max and min, and the frequency and intensity of heatwaves) do not follow a latitudinal gradient. There are important discontinuities in temperature regimes across latitude in this region, that are not taken into account, and could explain (at least in part) the lack of “trends” observed in cover, resistance or resilience. The wording when referring to trends (or the absence of it) sometime seems biased toward the willingness to find specific trends. For example: p17 “the interannual oscillatory nature of the canopy….likely contributed to the lack of a negative trend.” I understand this as the author failing to detect an existing negative trend. This would need rephrasing, in my opinion.

Another point, that is not really a flaw but should be discussed somewhere: the analysis of resistance, resilience and trends all rely on kelp canopy extension and its dynamics. First, it represents only part of the population, the one that reaches the surface. I do not know if there was any sort of validation with in situ observations; if so, they should be reported, if not, a statement of the possible underestimation of population dynamics should be made. Fig. 7 further illustrates this point: it shows that surface canopy area is not indicative of population persistence or resistance (and therefore, of resilience) as it reaches values close to 0 each year (during winters). It seems therefore that the inferences are more about resistance and resilience of biomass production than about population dynamics. This needs to be clarified as it can lead to misinterpretations

Minor observations:

The description of results is often highly qualitative and lacks of quantitative precisions: “a prolonged period of low canopy” (p11); “there was a considerable local scale variability” (p12), and other similar cases. Authors should report numbers instead of such qualitative and highly personal appreciations of the results.

The authors make emphasis on the use of Kelpwatch by non-specialist readers such as the stakeholders, decision makers, etc. While the visualisation tool is indeed highly valuable, some precautions must be considered in the light of possible misinterpretations about population dynamics (discussed above). Moreover, the patterns and trends described in this study are not fully understood (the authors provide limited explanations in terms of demography, population persistence, biogeography), which creates another risk of misinterpretation. Such cautionary principles should be made explicit if the main objective is to reach decision makers.

Fig 2 and 3 seem largely overlapping in scope and should be fused.

Reviewer #2: Overall comments:

The authors present a visualization tool called Kelpwatch to monitor changes and trends in kelp forests at different spatial scales between the Western United States to Mexico, using a satellite-derived kelp canopy dataset. They also present three levels of spatial/temporal analysis to determine kelp resistance and resilience; firstly, covering six regions along the complete time series, followed by the same regions after the event known as ‘the Blob’, and finally, a smaller area covering the Monterey Peninsula. The cases are well documented by previous analysis, except for the declines in Monterey that are identified thanks to this tool.

Despite the fact that both goals of the paper are achieved, a) to present Kelpwatch as a visualization tool; b) to present case studies of resistance and resilience, both can be improved in general and specific terms. For a), the authors may consider that the potential readers could be better guided to use this tool if the figures are improved in the article. Some geographical names mentioned in the text are missing from the maps. The boundaries of the six regions are indicated only in the supplementary figure 2. The readability of the figures by color-blinded people, or in hard-copy versions, can increase if the authors change the color palette and include symbols (for the scatterplot in figure 5). In other words, I encourage the authors to stimulate the use of the tool by increasing the accessibility of the article.

On the second goal, the authors are presenting cases of resilience and resistance through a series of generalized least squares regression models, where each p-value indicates the level of significance. Although I do not disagree with that method, it is interesting to see that all the plots show strong dynamism, with peaks and lows that not necessarily can be correlated with events like the Blob and, except for Oregon, most of the cells show no significant trend. I think that conceptual models of ecological resilience, i.e., hysteresis, regime shifts or alternate stable states (see Pelletier 2020 for definitions) could help to improve the analysis of the trends observed, even more considering the enormous amount of space and time covered by this research.

Specific comments:

- Page 4, first paragraph: you may say square meters or kilometres (instead of kms)

- Last paragraph of the introduction, page 5; why the local analysis in Monterey Peninsula only? This requires a short explanation, maybe moving the first paragraph starting on “the area surrounding the Monterey Peninsula exhibited an overall loss…’ on page 10.

- Results, pages 11-12: please ensure that the text goes in the same order as the plots in figure 2. This figure should have letters to label each plot (2a to 2f)

- Figure 3. Please indicate the boundaries of the six regions in the map, with their names and their plots (maybe making a fusion with Figure 2), including the geographical names mentioned in the text, i.e., Channel Islands. Please include a vertical bar to indicate the Blob years.

- Results, page 12. The majority of the cells show no trend, but this is not mentioned in the text. Why?

- Table 1. For consistency, modify the names ‘Baja Norte’ and ‘Baja Sur’ to ‘Baja California Norte’ and ‘Baja California Sur’

- Figures 3,4, and S4: the color palettes aren’t great for color-blinded people or for hard-copy versions. Please consider changing the palette to Batlow (Crameri et al., 2020) or similar.

- Figure 5: include symbols to differentiate kelp species.

- Figure 6 is a bit transparent, and the numbers 1,2, and 3 are almost invisible. Please increase the visibility of the images.

- Discussion, pages 16-17: Despite the fact that Oregon had 5 cells showing negative trends, it also showed the highest resistance to the Blob. This is not discussed and I’d like to see a better explanation of these contrasting results.

- Discussion, pages 17-18: I wonder what is an ideal condition for a kelp forest; if the cold phase of a marine oscillation like La Nina is optimal, the lower area in the Blob years may be the optimal area given the higher temperatures. If that is the case, the regression model could be complemented with a more conceptual model to include these natural adaptive oscillations, to get a better understanding of what is optimal or suboptimal resistance and resilience.

References:

Pelletier, M. C., Ebersole, J., Mulvaney, K., Rashleigh, B., Gutierrez, M. N., Chintala, M., ... & Lane, C. (2020). Resilience of aquatic systems: Review and management implications. Aquatic sciences, 82(2), 1-25.

Crameri, F., Shephard, G.E. & Heron, P.J. The misuse of colour in science communication. Nat Commun 11, 5444 (2020). https://doi.org/10.1038/s41467-020-19160-7

6. PLOS authors have the option to publish the peer review history of their article (what does this mean?). If published, this will include your full peer review and any attached files.

Reviewer #1: No

Reviewer #2: No

---

## [Author Response · Author response to Decision Letter 0]

14 Jan 2023

Thank you for your reviews of this manuscript. Please see response to reviewers document.

---

## [Decision Letter · Decision Letter 1]

6 Mar 2023

Kelpwatch: A new visualization and analysis tool to explore kelp canopy dynamics reveals variable response to and recovery from marine heatwaves

PONE-D-22-18739R1

Dear Dr. Bell,

We’re pleased to inform you that your manuscript has been judged scientifically suitable for publication and will be formally accepted for publication once it meets all outstanding technical requirements.

Kind regards,

Alejandro Pérez-Matus

Academic Editor

PLOS ONE

Additional Editor Comments (optional):

Dear Authors

I have sent the revised version of your manuscript "Kelpwatch: A new visualization and analysis tool to explore kelp canopy dynamics reveals variable response to and recovery from marine heatwaves" to the external reviewers and together we have made the decision that you have taken into account the opinion and suggestions raised by two reviewers who evaluated your manuscript. Consequently I have decided to accept your article in this version. Thank you for solving all the points issued by the reviewers and use PLOS ONE as channel of your research.

Reviewers' comments:

Reviewer's Responses to Questions

**Comments to the Author**

1. If the authors have adequately addressed your comments raised in a previous round of review and you feel that this manuscript is now acceptable for publication, you may indicate that here to bypass the “Comments to the Author” section, enter your conflict of interest statement in the “Confidential to Editor” section, and submit your "Accept" recommendation.

Reviewer #2: All comments have been addressed

2. Is the manuscript technically sound, and do the data support the conclusions?

Reviewer #2: Yes

3. Has the statistical analysis been performed appropriately and rigorously? 

Reviewer #2: Yes

4. Have the authors made all data underlying the findings in their manuscript fully available?

Reviewer #2: Yes

5. Is the manuscript presented in an intelligible fashion and written in standard English?

Reviewer #2: Yes

6. Review Comments to the Author

Reviewer #2: (No Response)

7. PLOS authors have the option to publish the peer review history of their article (what does this mean?). If published, this will include your full peer review and any attached files.

Reviewer #2: No

---

## [Editor Report · Acceptance letter]

13 Mar 2023

PONE-D-22-18739R1 

Kelpwatch: A new visualization and analysis tool to explore kelp canopy dynamics reveals variable response to and recovery from marine heatwaves 

Dear Dr. Bell:

I'm pleased to inform you that your manuscript has been deemed suitable for publication in PLOS ONE. Congratulations! Your manuscript is now with our production department. 

Kind regards, 

on behalf of

Dr. Alejandro Pérez-Matus 

Academic Editor

PLOS ONE